# OFDM System Design for Measured Ultrasonic Underwater Channels

**DOI:** 10.3390/s22155703

**Published:** 2022-07-29

**Authors:** Pablo Cobacho-Ruiz, Francisco Javier Cañete, Eduardo Martos-Naya, Unai Fernández-Plazaola

**Affiliations:** Communications and Signal Processing Lab, Telecommunication Research Institute (TELMA), ETS Ingeniería de Telecomunicación, Universidad de Málaga, 29010 Málaga, Spain; francis@ic.uma.es (F.J.C.); eduardo@ic.uma.es (E.M.-N.); unai@ic.uma.es (U.F.-P.)

**Keywords:** underwater acoustic communications, OFDM, linear distortion, coherent modulation, differential modulation, channel equalization, multipath propagation, channel sounding, synchronization

## Abstract

In this paper, we present the development of a multicarrier modulation system of low complexity for broadband underwater acoustic communications (UAC), whose frequency band is located in the ultrasonic range, specifically between 32 kHz and 128 kHz. Underwater acoustic channels are recognized among the most hostile communication channels due to their strong time and frequency selectivity and, hence, the design of high-performance systems is a challenge that is difficult to resolve at the present time with state-of-art technology. The aim of the proposed system is to reach a reasonable bit rate, between 40 and 50 Kbps, over these channels that allows, for instance, the transmission of video signals of limited quality. We describe an orthogonal frequency division multiplexing (OFDM) modem prototype with a parameter setting and design specifically adapted to the channel nature. For this purpose, actual measurements carried out at the Mediterranean sea, on shallow waters, have been used to evaluate the system performance and to optimize the design. A discussion on several modulations and OFDM configurations is presented that leads to the selection of differential and non-differential quadri-phase shift keying (QPSK) as good candidates depending on synchronization capabilities.

## 1. Introduction

The interest in Underwater Acoustic Communications (UAC) has increased in recent decades by the hand of technological progress, making possible the development of new activities and boosting other existing ones. Among these, we can mention the monitoring of environmental parameters, observation of marine fauna, aquaculture, detection of mineral resources, deployment of submarine cables, military applications, recreational activities, etc. Although wired solutions can be simpler for UAC systems, from a technological point of view, they can be too restrictive for many applications, while wireless communication systems are more versatile.

Acoustic waves suffer less attenuation in water that electromagnetic waves, both in the radio frequency and the optical range, which makes UAC the best option for many applications [1]. However, signals transmission through UAC channels is a technological challenge since there is an important signal degradation that is difficult to compensate for [2]. Due to this and the high bandwidth of our system, no large coverage can be achieved in UAC systems; however, this is advantageous because reduces the impact on underwater fauna [3,4].

Depending on the depth of communication, UAC channels can be divided into two main types: deep waters, when the communication takes place at a depth of more than 100 m, and shallow waters, for lower depths. Deep-water communications are generally conducted in a vertical direction, perpendicular to the water surface. However, in shallow waters, communications are usually carried out in a direction parallel to the water surface and the transmitted signals suffer from multipath phenomena due to reflections on the surface and seabed, leading to a notable signal distortion. This work is focused on this type of UAC channel.

UAC channels can be modeled as a class of stochastic communication channels, which are frequency and time selective or, in the dual domain, with significant delay spread and Doppler spread [5,6]. The propagation mechanism in this channel is characterized by four main factors:High attenuation. Mainly due to absorption and scattering, which result in attenuation that increases with frequency and distance, limiting the effective bandwidth of these channels to tens of kHz for distances up to few km.Noise. Caused mainly by environmental components due to water movement, tides and turbulences, breaking waves, wind, snapping shrimp [7], rain, boats, etc. There are reasonably good models of UAC noise [8]. This type of noise is not white but colored, exhibiting a decay with increasing frequency, which results from sea attenuation with frequency acting as a low-pass filter for distant noise sources.Multipath propagation. The multiple reflections on the surface and seabed, coupled with the low propagation velocity of acoustic waves in water, result in very long, multi-component impulse responses. This strong time dispersion results in high frequency selectivity. The response of UAC channels has traditionally been approached from the field of physics [9,10], and developed models, from a physical description, contains the different paths (or rays) that the signal follows after the reflections, and scattering it can be seen in the undersea environment. These models are often referred to as geometric models and there are several proposals of this kind [1,11].Time variation. The channel has an impulse response in which each physical path presents a variant behavior [12] resulting from time variations, such as transmitter and receiver movements, waves in the surface, tides, turbulences, changes in the speed of sound that are deep sea-dependent and other phenomena [13]. Hence, it becomes a linear time-varying (LTV) system that introduces, in addition to the time dispersion, a frequency dispersion or Doppler shift. Moreover, due to the low speed of sound propagation in water, changes in the channel response are very relevant and the under-spread condition of the LTV channel [14] is compromised, since the coherence time can be of the same order of magnitude as the effective impulse response duration.

The aim of this work is to pave the way for a system prototype able to transmit medium data rate signals, such as the ones for low-quality video, with a modest implementation complexity. For that purpose, we employ broadband in the ultrasonic range to compensate for the use of simple modulation formats with low spectral efficiency and still reach a reasonable data rate. It must be noted that, although the total bandwidth used in the system under consideration here may seem low, it is broadband in fact because the signal bandwidth fairly exceeds the channel coherence bandwidth, and it is similar to the center frequency. The selection of orthogonal frequency division multiplexing (OFDM) for our design supports the objective of obtaining a simple and broadband system that allows for more efficient frequency equalization.

In this regard, there are previous works that focus on UAC systems using these modulations both from a more theoretical perspective [15,16] or from a signal processing point of view but for more sophisticated receiver designs that the one we aim at here. Among the latter, we would like to mention: Ref. [17], where a zero-padded OFDM system is proposed with ad hoc intercarrier interference (ICI) compensation techniques; Ref. [18], where a multiple-input multiple-output (MIMO) OFDM system is described with iterative demodulation and decoding; Ref. [19], where also MIMO OFDM is applied to UAC with adaptive channel estimation; Ref. [20], which explores the possibilities of adaptive modulation for these kind of UAC systems but with a radio frequency feedback channel; and [21], which is a non-conventional approach to OFDM that presents a demodulation with a few fast Fourier transforms (FFT) and a linear combiner. Our system is wider bandwidth and higher frequency than these other solutions and this makes these more sophisticated techniques more difficult to apply and would offer worse performance.

We consider both phase shift keying (PSK) and differential PSK (DPSK), since the latter avoids the need for channel estimation and the use of pilots, supporting our goal of preserving bandwidth efficiency. Moreover, we study two kinds of DPSK: differential in time (DT-PSK), between consecutive OFDM symbols and differential in frequency, between subsequent sub-carriers (DF-PSK) to face the dual time- and frequency-selective nature of the channel.

On the other hand, we choose an implementation of the OFDM signal directly in the passband, as is the usual approach in discrete-multitone (DMT) wired systems, which avoids the need of carrier frequency synchronization that is naturally engaged to the sampling frequency. This choice is also based on the fact that we face a Doppler shift that is very inhomogeneous for the subcarriers in the whole band (the bandwidth used is several orders of magnitude larger than the coherence bandwidth [22,23,24]), which impedes our ability to make a carrier frequency offset (CFO) compensation like in other works focusing on narrower-band UAC. In [25], an interesting proposal of a Doppler-tolerant receiver with signal resampling for an UAC system is presented, but it works in a smaller bandwidth and for single-carrier signals.

In order to deal with these doubly selective UAC channels, we propose the use of a large number of carriers which increases, for a given bandwidth, both the spectral efficiency, the ratio between useful OFDM symbol part and the cyclic prefix (CP) one, and the coherence between carriers (with a smaller sub-carrier spacing) that helps differential encoding in frequency. However, this choice is not exempt of problems, because it enlarges the OFDM symbol duration and makes the system more susceptible to the channel variation. This is because the symbol interval may approach or even exceed the channel coherence time, which loses orthogonality between carriers. In that situation, the ICI degrades both coherent and differentially coherent detection [21].

In this paper, a series of system configurations will be proposed, whose performance, assessed by means of simulations, is satisfactory. Simulations have been carried out by testing the system over a selected set of eight channels obtained from measurements in different trials in the Mediterranean Sea that we consider representative of the expected behavior of shallow-water UAC channels.

These measurements were taken by the authors of this article in different measurement campaigns carried out on different dates (2017, 2018 and 2019) and in different locations in the Mediterranean Sea (Cartagena and Malaga). Both boats were separated distances ranging from approximately 50 to 400 m and the observed sea depth along the channels spans from 19 to 34 m. On the other hand, the type of seabed in both locations was sandy with some sparse rocks.

We take into account practical issues in our configuration selection, such as the impact of imperfect channel estimation, the rate loss due to pilots insertion, the sensitivity to ICI caused by the channel time variation, the residual intersymbol interference (ISI) when the CP duration is less than that of the impulse response, etc. We also study design alternatives such as a less-dense pilot distribution to test the tradeoff between the channel estimation quality and the reduction of the ICI among pilots, or the use of a very simple but suboptimal synchronization scheme.

The remainder of this paper is organized as follows. Section 2 presents the selected scenarios and the equipment used in the measurements. Section 3 provides an overview of the proposed system model; also, the OFDM system design is discussed, detailing the most important aspects of the transmitter, receiver and encoder. In Section 4, we describe some relevant details of the simulated system implementation to test the performance of the proposed configurations, whose results will be discussed in Section 5. Finally, some conclusions are summarized in the last section.

## 2. Underwater Measurements Description

We have carried out several measurement campaigns in shallow water of the Mediterranean Sea, in order to characterize the UAC channel both in narrowband [26] and in a wider band, at ultrasonic frequencies centered at 80 kHz with a width of 96 kHz [22]. Two boats are employed to locate the transmitter and receiver, which were separated by a range of distances between approximately 50 and 400 m, the water depth varied between 19 and 34 m and the sea bottom type was sandy with some rocks. The transducers are both at the transmitter end, (the projector), and at the receiver end (the hydrophone) and submerged at a fixed depth of 6 m from the water’s surface.

The measurement equipment is shown in the photograph in Figure 1 and a simplified schematic of the hardware used for the measurements is shown in Figure 2. In terms of hardware, the transmitter end includes a laptop (for control, storage and signal processing), an acquisition module with a resolution of 16 bits and a sampling rate of 500 kHz, which was used as D/A converter; a power amplifier and a projector. On the other hand, the receiver end consists of: a second laptop; an acquisition module, with a sampling frequency of 500 kHz, which was used as A/D converter; a preamplifier; and a low-noise hydrophone. Specific software is used to obtain long-term measurements in an automated way, using broadband sounding signals that are post-processed to characterize the UAC channel.

Figure 3a shows an example of the measured impulse response of a time-varying UAC channel, in which the time variation of the main path or Line-of-Sight (LOS) component (with the lowest delay) followed by the other paths (due to the reflections in the surface and in the seabed) can be seen. This estimated UAC channel response was obtained from the measurement campaign discussed above. Specifically, the transmitter–receiver separation for the channel in Figure 3a was approximately 243 m.

To obtain the aligned response shown in Figure 3b, we first estimated the initial delay variation from the unaligned impulse response. Then, we deduced the resampling frequency of the received signal that compensates for that variation to estimate the aligned channel response [22]. This compensation technique would be equivalent to the synchronization required to obtain the best performance in an OFDM system. Note that in Figure 3b, the echo structure of the impulse response remains fairly invariant throughout the measurement.

As mentioned above, eight UAC channel responses have been estimated in order to test the performance of the proposed OFDM system. The performance that can be obtained with these channels does not depend exclusively on the separation between transmitter and receiver, but there are more influential factors such as the movement of the boats or the sea conditions. In this regard, measurement campaigns use a large set of sounding signals that extend the measurement of a particular channel by up to 15 min. For each new channel measurement, the hydrophone configuration and boats’ locations are changed.

These eight channels have been selected because we consider them representative of the set of those measured in different tests, sites and dates. Additionally, we have sorted them from best to worst according to the performance obtained with them, which are discussed in Section 5. Table 1 shows the transmitter–receiver separation for the eight measured channels.

Note that channel 8, despite having the smallest transmitter–receiver separation, generally gives the worst results, because the movement experienced by the boats at that measurement was greater than usual.

## 3. System Model

This section briefly describes the system model and the most important aspects of the OFDM system design: DMT generation of the OFDM signals, OFDM transmitter and receiver, main OFDM parameters, such as the number of FFT points or CP length; in addition to channel coding and interleaving. Finally, the data rates obtained with the different configurations are explained. As can be seen in the diagram in Figure 4, the system is composed of the following elements:Channel encoder. This element is responsible for the coding and interleaving of the information bits using convolutional coding.OFDM transmitter. This receives the encoded bits, generates the modulation symbols and creates the structure of the bandpass OFDM signal, ready to be transmitted.UAC channel. This incorporates the response of a intra-symbol time-varying channel and additive noise, both estimated from actual measurements.OFDM receiver. Receives the OFDM signal by means of synchronization and windowing techniques, compensates for the channel response, detects the modulation symbols and estimates the received coded bits.Channel decoder. This reconstructs the original bits of information using the Viterbi’s algorithm trying to reduce the bit error rate (BER).

### 3.1. OFDM Transmitter Based on a DMT Approach

The OFDM transmitter is responsible for generating and transmitting the complete OFDM signal from the coded bits. Regarding its structure, it is composed of several elements shown in Figure 5:M-PSK symbol mapping. This channel, due to its time-varying nature, introduces significant variations in the phase and amplitude of the received signal. For this reason we have chosen simple and robust modulations such as PSK and D-PSK, which are discussed in Section 3.3.Subcarrier selection and pilot insertion. Each generated PSK or D-PSK symbol is placed on its corresponding OFDM carrier and symbol as explained below. In case of using PSK modulations, the pilot carriers consist of known QPSK symbols and are added at their corresponding locations for channel estimation at the receiver.Cyclic prefix insertion. Due to the typically long time duration of the measured UAC channel response, we have chosen a long CP with respect to other OFDM systems to reduce ISI and ICI. An interesting alternative to reduce the complexity of the system would be the use of Zero-Padding OFDM; however, this technique hardly reduces the ICI [27].

The frequency conversion process of this OFDM transmitter is based on the DMT technique, which does not require an ulterior process of up-conversion in the analog domain. This is possible because the sampling frequency of our system, $f_{s}=500$ kHz, is above twice the maximum frequency of the signal in the working band, which lies between $f_{a}=32$ kHz and $f_{b}=128$ kHz.

The generated signal, X(k), represents the OFDM symbol in the frequency domain and is shown in Figure 6. The main idea is to place the PSK symbols to be transmitted on the appropriate active carriers to directly obtain the bandpass signal. Let *N* denote the number of FFT points; $N_{a}$ the number of active carriers (carriers on which the PSK symbols shall be placed); *k* the carrier index (k∈[−N/2,N/2−1]) and A(l) the vector of PSK symbols (with $l=1,2,\ldots ,N_{a}$).

X(k) vector is created with the $N_{a}$ PSK symbols and their $N_{a}$ complex conjugate values (which is denoted by $A^{*}$) in the corresponding positions to obtain the transmitted signal, in the work band between $f_{a}$ and $f_{b}$. Finally, the real signal in the time domain is obtained by applying the inverse fast Fourier transform (IFFT).

In addition, this implementation allows for simultaneous symbol and carrier synchronization as discussed in Section 3.2, since the sampling frequency determines carrier frequency.

### 3.2. OFDM Receiver

After the transmitter and the UAC channel, we find the receiver block, which is composed of the following elements (see Figure 7):Windowing and CP detection. This selects the time samples of the incoming OFDM symbol with a rectangular window and deletes the part corresponding to the CP.Synchronization. This is responsible for carrier and symbol synchronization. The strong channel variation causes an expansion or compression of the signal in time, which must be compensated by resampling as described in [22]. This represents a sophisticated synchronization technique that is assumed in the remainder of this section. Possibly, the use of these techniques in a prototype can be challenging. However, simpler synchronization techniques have been tried that offer reduced performance but still make system operation possible as will be discussed in Section 5.3.Subcarrier Selection. Extracts the data and pilot symbols (in case of PSK modulation) from their corresponding subcarriers.Channel Estimation. This is responsible for estimating the channel from the information obtained from the pilot carriers. A noisy estimate of the channel frequency response at the pilot frequency is used; this response is then interpolated for the data carriers.Frequency Equalizer (FEQ). If a PSK modulation is selected, the FEQ compensates for the channel effects as far as possible.M-PSK Detector. This block obtains the transmitted symbols to obtain the information bits.

### 3.3. Main OFDM Parameters

Regarding the main parameters used in the OFDM system, as shown in Table 2, the sampling frequency of the system is 500 kHz and FFTs of 8192 and 4096 points have been tested. The reason for this is to deal with the low-coherence bandwidth of the UAC channel and to somehow improve the spectral efficiency that is rather compromised by the long CP of the system. In this regard, two values for the number of samples of the CP have been used: 4096 and 2000. With these two values, the CP duration is larger than the effective length of the channel impulse response, which we define as the one that comprises the 98% of its energy.

In terms of the frequency band used, as mentioned above, this paper deals with signals working in the ultrasonic band, specifically in the band between 32 and 128 kHz. The number of active carriers allows the configuration of the bandwidth used: when the number of FFT points is N=8192, the number of active carriers is $N_{a}=1576$. Similarly, to maintain the same bandwidth, when N=4096, half the number of active carriers is used, $N_{a}=788$ (see Figure 6).

Regarding modulations, we have proposed the following for this system: PSK, DT-PSK and DF-PSK. In the case of PSK modulation, in order to estimate the channel, pilots are inserted alternately with the data carriers, using half of the active carriers as data and the other half as pilots. On the other hand, differential modulations do not require pilot carriers as, by nature, they compensate for the phase shift drift experienced by the data carriers. Thus, all active carriers can be used as data carriers, with the resulting increase in bit rate.

Specifically, the PSK modulations selected are Binary Phase Shift Keying (BPSK) and Quadrature Phase Shift Keying (QPSK). On the other hand, the proposed DT-PSK modulations are Differential Time BPSK (DT-BPSK), Differential Time QPSK (DT-QPSK), Differential Frequency BPSK (DF-BPSK) and Differential Frequency QPSK (DF-QPSK).

For DT-PSK modulation, all PSK symbols carried by the first OFDM symbol are used as reference for modulation. On the other hand, for DF-PSK modulation, the PSK symbol corresponding to the first carrier of each OFDM symbol is used as the reference.

Finally, as observed in Table 3, in order to obtain a system that offers good performance, the following three sets of N and M parameters have been selected based on the tradeoff between channel variation (sufficiently short symbol) and ISI-ICI reduction (sufficiently long CP):

### 3.4. Channel Encoding and Interleaving

With the aim of reducing the BER of the system, a non-recursive and non-systematic convolutional encoder has been selected, whose coding rate is 1/2 and whose diagram is shown in Figure 8.

On the other hand, error burst in frequency due to fading subcarriers can drastically reduce the performance of channel coding. In addition, the Signal-to-Noise and Distortion Ratio (SNDR) in the whole band is not uniform because both the noise is not white and the higher subcarriers are more affected by the ICI, since the phase-shift between successive samples is much larger than in lower subcarriers. To mitigate these effects, we have implemented interleaving with a depth of two OFDM symbols, by filling a matrix by rows and reading it by columns before symbol mapping at the transmitter and changing the role of rows and columns in the de-interleaving applied after detecting symbols at the receiver. Hence, the interleaving helps to make the performance of the active subcarriers more homogeneous. The size of the interleaver matrix is set to extend for two OFDM symbols.

### 3.5. Achieved Data Rates

Once the OFDM system design has been discussed, we show in Figure 9 the bit rates obtained for the different modulations and configurations of *N* and *M* parameters, considering the coding rate detailed in Section 3.4. Note that PSK modulations, unlike D-PSK modulations, need to use the carrier pattern in which half of the carriers are pilots. For that reason, PSK modulations have half the bit rate of D-PSK modulations. This is not strictly correct in the case of DF-PSK modulation, since the first active carrier of each symbol (with the lowest frequency) is used as a reference for demodulation at the receiver; this implies a bit rate loss of less than 0.1% with respect to the case of DT-PSK modulation.

## 4. Numerical Simulations

This section describes some relevant details of the system implementation that has been simulated, such as the channel variation and estimation (in case of PSK modulation) and UAC noise modelling.

### 4.1. Channel Estimation and Frequency Equalization

In the case of PSK modulation, the channel must be estimated to compensate for the impairments introduced by the channel. In the simulations, a pilot-based channel estimator has been implemented, whose information is used as a reference for the channel estimator. In channels with fading and fast variation, the pilot information must be transmitted as continuously as possible. We have adopted a pilot pattern which interleaves the pilot carriers with the data carriers, so that half of the carriers are used as pilots. In this way, the channel is estimated at the position of the pilots and the frequency response of the channel between pilots is estimated by means of interpolation. Since the use of pilots reduces the binary regime, there is a tradeoff between good channel estimation and bit rate.

In terms of channel compensation, since distortion dominates over noise on these measured UAC channels, a modified Zero-Forcing (ZF) solution has been implemented in order to avoid excessive noise enhancement when the attenuation of a given carrier is very high. Once the frequency domain channel response has been estimated, the FEQ for the data carriers is calculated as the inverse of the frequency domain channel response:(1)$$FEQ\left[k\right]=1/(H_{est}\left[k\right]+\delta ),$$
where $H_{est}\left[k\right]$ is the estimated frequency response at the data carrier frequency and δ is a small constant offset that prevents noise enhancement on very attenuated carriers. So, the compensated received signal is:(2)$$Y_{eq}=FEQ\left[k\right]\cdot \left(Y\left[k\right]+N\left[k\right]\right),$$
where Y[k] is the received signal that exhibit channel degradation (distortion, ISI, ICI, etc.) and N[k] is the UAC noise registered at the *k*-th sub-carrier.

### 4.2. Noise Modelling

In order to introduce underwater noise into the simulation, noise samples in the time domain have been synthesized from a UAC noise power spectral density (PSD) as shown in Figure 10. This spectral density has been obtained from the measurement campaign mentioned in Section 2, applying a processing to estimate the stationary spectral density, by eliminating some randomly fluctuating narrowband components.

The aim is to generate a colored Gaussian noise with a PSD similar to that obtained in the measurement campaign, with a decay in frequency, by means of a moving average process (MA). This process consists of coloring Additive white Gaussian noise (AWGN) by filtering.
(3)$$S_{n}\left(f\right)=|H_{n}{\left(f\right)|}^{2}\cdot S_{w}\left(f\right)={\left|H_{n}\left(f\right)\right|}^{2},$$
where, for simplicity, $S_{w}\left(f\right)=1\left[W/Hz\right]$ and $H_{n}\left(f\right)$ is the frequency response of the shaping filter, which has the shape of the measured UAC noise PSD, $S_{meas}\left(f\right)$ derived from
(4)$$H_{n}\left(f\right)=\sqrt{S_{meas}\left(f\right)}.$$

The noise simulation is done by means of time domain filtering, where the impulse response of the shaping filter is calculated from the inverse Fourier transform of the frequency response.

### 4.3. Implementation of Channel Time-Varying Behavior

For the implementation of this time variation of the channel, a simple method, derived from the schematic in Figure 11, has been implemented in which a channel variant filter is constructed by means of a zero-order hold interpolation [28].

First, note that the time resolution with which the channel has been measured, $T_{m}$, is approximately the time duration of an OFDM symbol according to the parameters in Table 2 for configuration 2. However, in order to obtain a more realistic simulation of the channel variation, we have performed a fourth-order interpolation of the channel impulse response so that the new time resolution is $T_{c}=T_{m}/4$. In this way, for configuration 2 the channel varies among four states within the same OFDM symbol, i.e., the *i*-th OFDM symbol suffers the effect of four impulse responses that are considered as time invariant (LTI). The longer the OFDM symbol length, the greater the number of times the channel varies within the same symbol (intra-symbol channel changes) as shown in Table 4.

As shown in Figure 11, the *i*-th OFDM block, $x_{i}\left(t\right)$, passes through this filter, simultaneously feeding all four phases and the selector permutes the output every $T_{c}$ seconds forming the output signal, $y_{i}\left(t\right)$. The blue block corresponds to the filter of the measured response, while the gray blocks are those of the responses obtained by interpolation. This scheme is repeated for each OFDM transmitted block.

Note that the measured channel responses have a duration of 1 min (as shown in Figure 3), allowing them to transmit more than 2000 OFDM symbols, with the longest symbol configuration presented in Table 2 and providing good resolution for the simulation.

## 5. System Performance Results

This section presents the results obtained from simulations of the proposed set of modulations over the eight channels measured. In addition, we have evaluated the impact of a less-dense carrier pattern and a simple synchronization technique to explore other design alternatives. Moreover, although SNR is not constant in practice, it has been found that noise is not the most limiting factor. Thus, in all simulations, the noise level has been set such that the SNR at the receiver input is approximately 20 dB.

### 5.1. Evaluation of Modulations Set

Firstly, the performance of the system for the six modulations and the three proposed *M* and *N* configurations have been evaluated by means of simulations. The BER obtained for the simulations of configuration 2 are shown in Figure 12, while the BER obtained for configurations 3 and 4 are shown in Figure 13 and Figure 14, respectively. Figure 12a shows the results in a heat map-like table while Figure 12b, Figure 13 and Figure 14 show the same results in a more compact and less detailed format.

As noted, configuration 2 generally offers the lowest BER values. Specifically, the modulations that offer the best performance are PSK and DT-PSK modulations obtaining similar results, although DT-PSK modulation offers more consistent performance across the different channels tested. In addition, it is important to remember, as previously mentioned, that D-PSK modulations (without pilot carriers) achieve twice the bit rate of PSK modulations, making them a better choice.

On the other hand, as shown in Figure 13, the PSK modulations of configuration 1 achieve better results than the other configurations due to the decrease of the frequency spacing between carriers, which allows a better channel response estimation. This smaller carrier spacing also favors the results for the DF-PSK modulations, reducing the effect of the channel frequency response variation. However, as the time duration of the OFDM symbol increases, the signal suffers a stronger channel variation, a consequence directly suffered by DT-PSK modulations, offering worse results, in general, than configurations 2 and 3.

Finally, one of the advantages of the reduced CP of the configuration 3 is that, by reducing the symbol duration, it suffers less from the temporal variation of the channel; however, it also minimizes to a lesser extent the effect of ISI and ICI, systematically reducing the performance in all simulations with respect to configuration 2, as can be seen in Figure 14.

Looking at the results from the point of view of the bit rate vs. BER tradeoff, Figure 15 shows the BER and the bit rates (the bit rate has been taken directly from Figure 9) achieved for the QPSK, DT-QPSK and DF-QPSK simulations together (taking into account the fact that BER values for QPSK modulations can be sufficiently low for our communication system and, in addition, the obtained bit rates are twice as high as for BPSK modulations, for simplicity the results of the latter are not presented in Figure 15). This figure gives us a broader idea of what will be the most suitable configuration. As can be seen, QPSK modulations achieve the lowest BER values. However, these modulations are less consistent in the set of all simulations, presenting in some cases very high BER values.

Although DT-QPSK modulation offers higher BER values than QPSK modulation, they are still values that allow adequate communication performance. The main advantage of this modulation over QPSK is that DT-QPSK allows communications at much higher bit rates. Finally, DF-QPSK modulation offers consistently worse BER values while obtaining the same bit rates.

In short, in DF-PSK modulations the frequency selectivity is considerably high, which makes it very difficult to adapt to this variation, obtaining BER values that are too high. On the other hand, simulations with DT-PSK modulation improve the results of the previous modulation, sometimes obtaining results similar to those obtained in PSK modulation with channel equalization. From these results we can draw several conclusions:Time variation is less abrupt (once the main path delay has been compensated by using the technique discussed in Section 2 and explained with Figure 3b) than frequency selectivity, allowing the use of DT-PSK to give better results than DF-PSK.PSK modulation with channel estimation and DT-PSK modulation have sufficiently low BER values for proper communication. However, DT-PSK does not require channel estimation and can use all active carriers as data carriers, thus doubling the bit rate of the communication and simplifying receiver design.

### 5.2. Study of a Less Dense Carrier Pattern

In this subsection, we study the effect of using a less-dense carrier pattern in all configurations under analysis, with the objective of assesing whether a larger separation between carriers provides some performance improvement due to a reduction of the residual ICI. In the proposed patterns, the carriers can be active (data or pilot carriers) or empty. Figure 16 shows the tested carrier patterns: in pattern A all carriers are active, using all of them as data carriers in case of D-PSK modulations or alternating between pilot and data carriers in PSK modulations. On the other hand, the new pattern B is a less-dense pattern, whose spectral separation between the carriers is four times larger. So far, all the above results have been obtained with pattern A and we now test pattern B.

Comparing the results obtained with pattern B for system configuration 2, shown in Figure 17a, with those obtained with pattern A, shown in Figure 12a, it is observed that the pattern B generally offers a worse performance. In the case of PSK modulations, the BER increases due to the channel estimation error, since the pilot carriers have a much larger spectral separation and, for the channel coherence bandwith under consideration, this degrades the FEQ behavior. There is a tradeoff between combating distortion due to time selectivity and distortion due to frequency selectivity, both in the form of ISI and ICI. In the case of DF-PSK modulation, the results are worse because the differential detection is not able to compensate for the increased phase differences, which are larger than in pattern A. The DT-PSK modulation outperforms in the case of pattern B, however, the results are still far inferior to those for pattern A.

In any case, when using pattern B, the best configuration is the first one, with N = 8192 and M = 4096 (see Figure 18), better than both configuration 2 (see Figure 17) and configuration 3 (see Figure 19), but it gives consistently worse results than pattern A. PSK and DF-PSK modulations become worse for configurations with a smaller number of FFT points because the spectral separation between pilot carriers is even larger.

As observed, the use of less-dense carrier patterns does not introduce a substantial improvement in the BER results, and, on the contrary, has the adverse effect of greatly reducing the bit rate.

### 5.3. Evaluation of a Simple Synchronization Technique

In this subsection, we analyze a simpler synchronization technique than the one that would be equivalent to the resampling used in [22], looking for a real-time and cost-effective implementation. (Note that, so far, all the above results have been obtained with the sophisticated synchronization technique.) This simplified synchronization technique consists of estimating the time instant at which the OFDM symbol starts in the received signal [29]. For this purpose, we adopt a classical approach that takes advantage of the structure of the OFDM symbol, specifically the fact that the CP is an exact copy of the last part of the OFDM symbol. This synchronization technique is based on the autocorrelation of the CP as shown in Figure 20 and is given by [30]:(5)$$r_{co}\left[n\right]=\sum_{i=0}^{M-1}r\left[n+i\right]\cdot r\left[n+i+N\right],$$
where r[n] is the received signal, *M* is the number of samples in CP and *N* is the number of FFT points (or the number of samples of the OFDM symbol excluding CP). The peaks of this function will correspond to the initial samples of each OFDM symbol received in the absence of significant noise and distortion. It is necessary to normalize this correlation metric by the energy of the signal in the window to avoid samples with more amplitude (but less correlation) causing a false maximum. Let R[n] denote the estimation of this instantaneous energy contained in an M-length window,
(6)$$R\left[n\right]=\sum_{i=0}^{M-1}{\left|r\left(n+i+N\right)\right|}^{2}.$$

Thus, the selected correlation-based synchronization function takes values between 0 and 1 and is defined as:(7)$$M\left[n\right]=\frac{|r_{co}{\left[n\right]|}^{2}}{R^{2}\left[n\right]}.$$

In order to analyze the performance of this simplified technique, we have carried out simulations on the measured channels with the different modulations proposed and for parameter setting 2, obtaining the BER results shown in Figure 21.

For D-PSK modulations, which do not make use of channel estimation, the synchronization technique employed is of great importance. As can be seen, these modulations significantly worsen the BER results with respect to the synchronization technique derived from [22] and whose results were shown in Figure 12. However, PSK modulations are still a good alternative in the case of employing this simple synchronization, allowing a BER of around $10^{-3}$ in most channels.

## 6. Conclusions

In this paper, ultrasonic UACs or shallow water have been analyzed by evaluating the performance of an OFDM-based system to operate over distances of a few hundred meters. A number of candidate modulation/detection scenarios, parameter settings and the use of coding/interleaving has been tested to substantially reduce BER and obtain sufficiently high bit rates. The frequency band has been set to 32–128 kHz, which offers a good region in terms of the UAC noise and provides a reasonable bandwidth.

We have compared the performance of the system for the different proposed configurations and modulations through simulations in terms of BER and bit rate over a selection of measured UAC channels. We have obtained a good performance with PSK and DT-PSK modulations, achieving bit rates above 64 Kbps in the case of DT-PSK modulations and BER below $10^{-4}$ in several channels. Specifically, we highlight the following settings:By using a sophisticated synchronization technique which would be equivalent to the resampling used in [22], the configuration with carrier pattern A, parameter setting 2 and for DT-PSK modulations gives consistent results throughout almost all channels, achieving bit rates above 48 Kbps, with a BER below $10^{-3}$ for most of the channels measured. In addition, this same configuration but using PSK modulations achieves bit rates above 24 Kbps, with BER values similar to those obtained with DT-PSK.By using the simple synchronization technique proposed in Section 5.3, with carrier pattern A, parameter setting #2 and PSK modulations, bit rates above 24 Kbps, with a BER below $10^{-3}$ can be achieved for most of the channels measured.

This system could be improved by designing a new synchronization algorithm not far from the simpler one in terms of complexity but closer to the sophisticated one in terms of performance. Further enhancement is possible by using more advanced coding techniques such as LDPC or Turbo codes and more sophisticated equalizers, in addition to the use of MIMO techniques, but our aim is to keep the implementation complexity at an affordable level to construct prototypes.

## Figures and Tables

**Figure 1 sensors-22-05703-f001:**
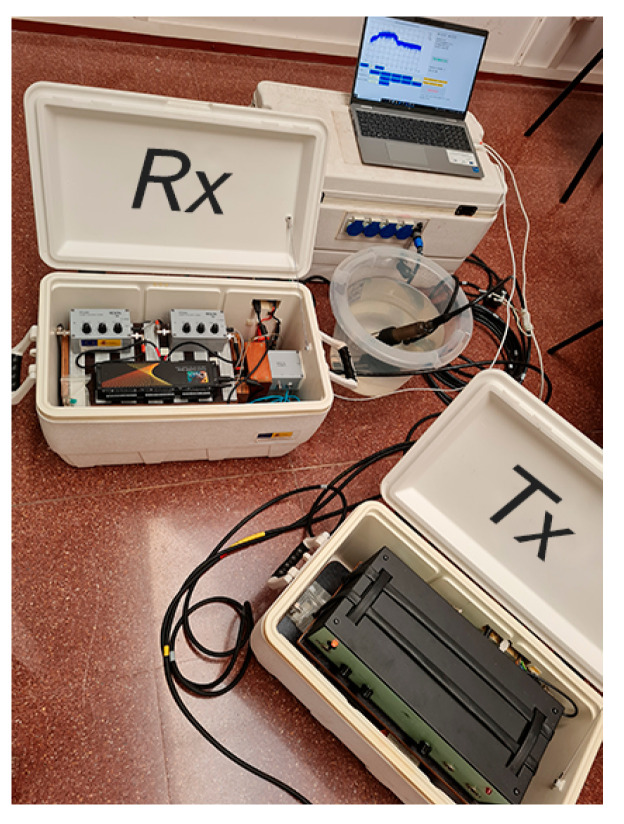
Measurement Equipment.

**Figure 2 sensors-22-05703-f002:**
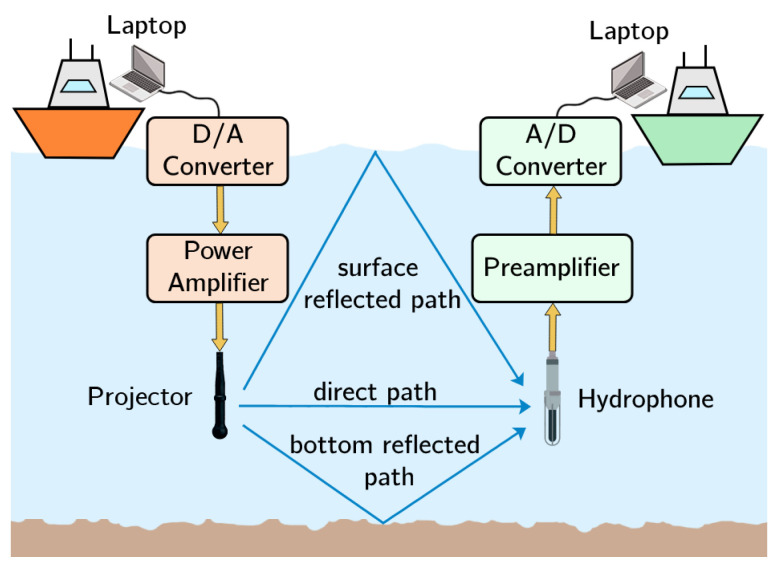
Diagram of the measurement scenario.

**Figure 3 sensors-22-05703-f003:**
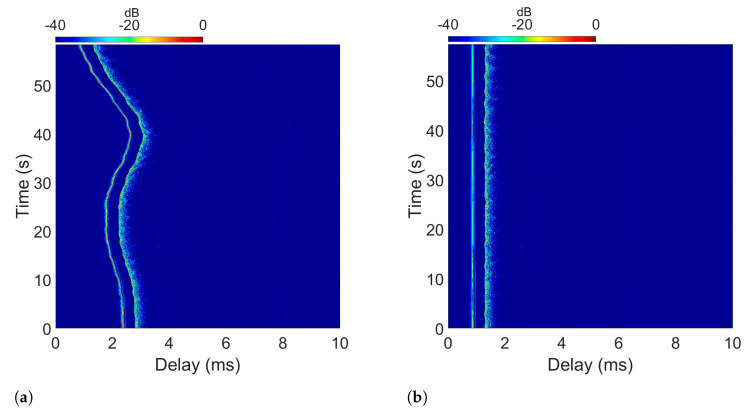
Estimated time-variant impulse response of one of the measured channels (**a**) before initial delay compensation |h(t;τ)| and (**b**) after initial delay compensation |h(t,τ)|.

**Figure 4 sensors-22-05703-f004:**
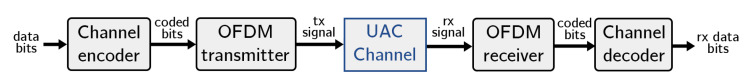
Block diagram of main elements in the system.

**Figure 5 sensors-22-05703-f005:**
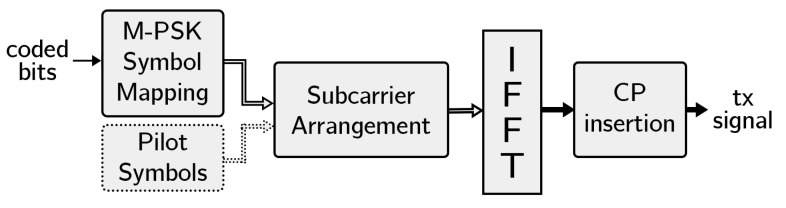
OFDM transmitter block diagram.

**Figure 6 sensors-22-05703-f006:**
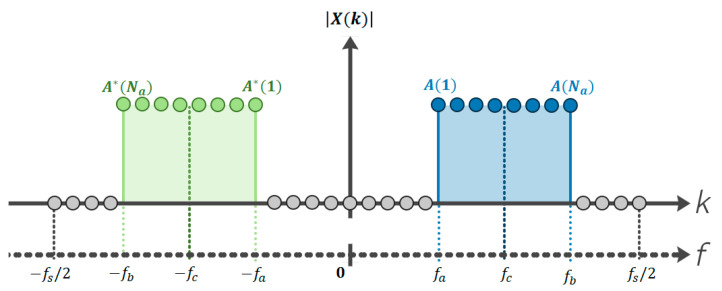
Mapping process for the signals at the input of the IFFT for the cases of a transmitter with DMT type bandpass signal generation.

**Figure 7 sensors-22-05703-f007:**
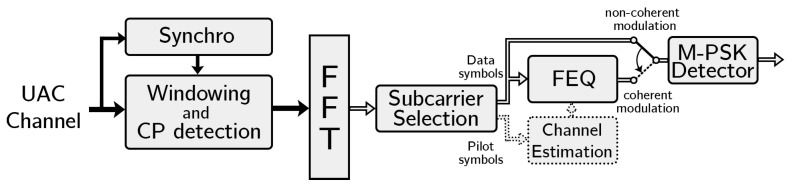
OFDM receiver block diagram.

**Figure 8 sensors-22-05703-f008:**
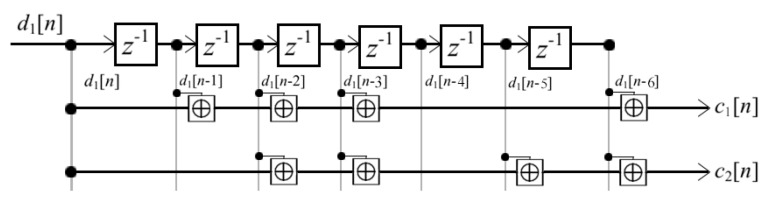
Diagram of non-recursive convolutional encoder implemented in the system.

**Figure 9 sensors-22-05703-f009:**
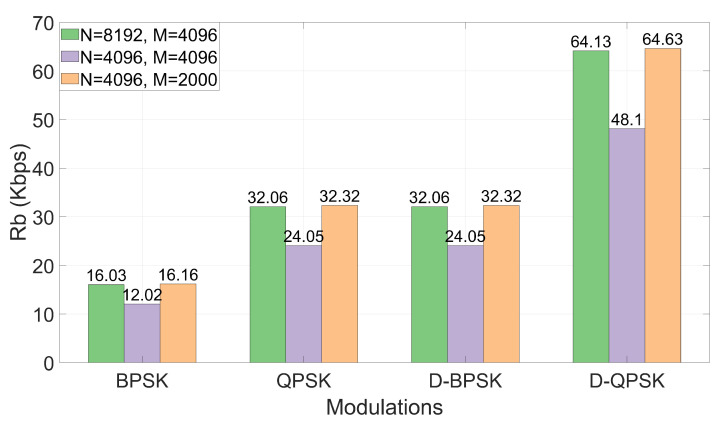
Bit rates obtained for the different configurations with channel coding applied.

**Figure 10 sensors-22-05703-f010:**
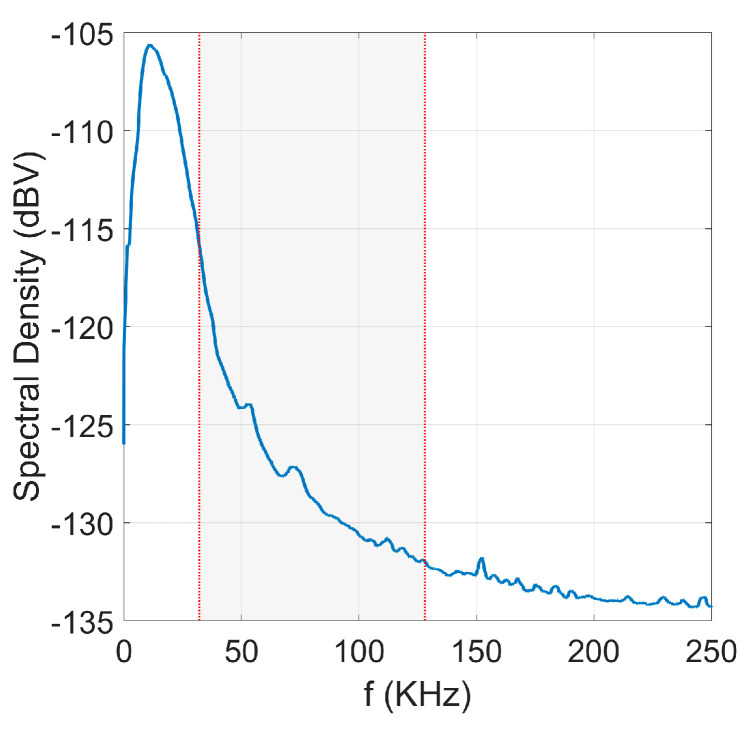
Spectral Density of measured UAC noise. The shadowed band correspond to the active subcarriers.

**Figure 11 sensors-22-05703-f011:**
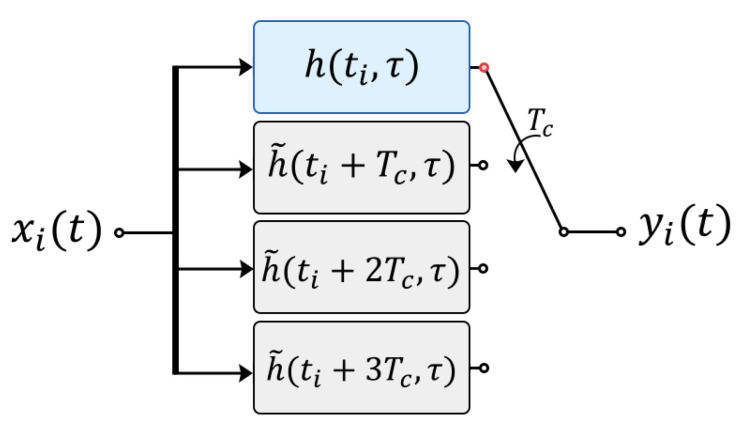
Implementation of the channel time-varying response in the *i*-th OFDM block by means of the zero-order hold interpolation of LTI polyphase filters.

**Figure 12 sensors-22-05703-f012:**
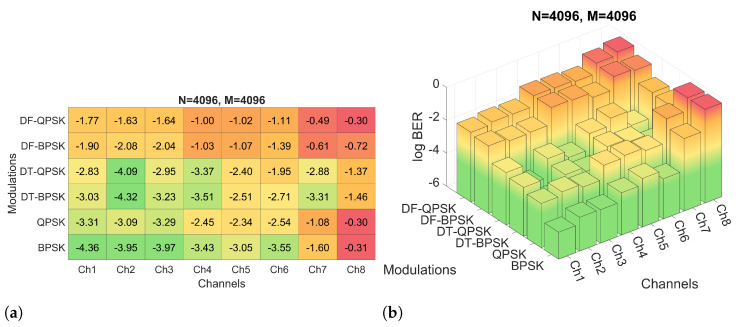
log BER for configuration 2 with different channels and modulations: (**a**) numerical heat matrix and (**b**) bar diagram.

**Figure 13 sensors-22-05703-f013:**
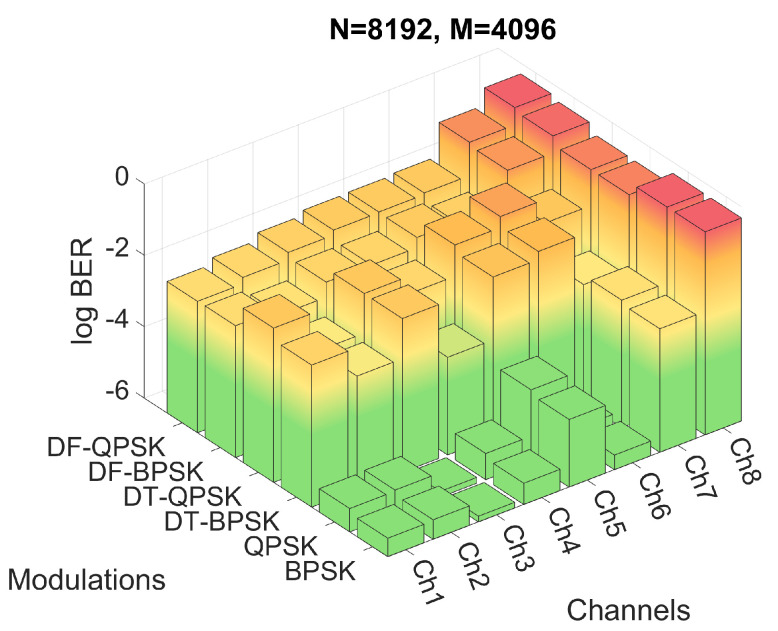
BER for configuration 1 with different channels and modulations.

**Figure 14 sensors-22-05703-f014:**
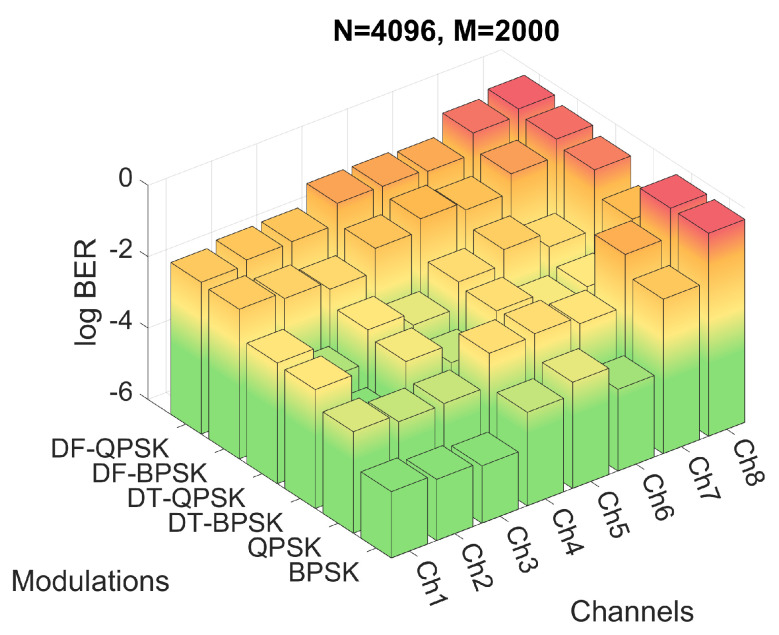
BER for configuration 3 with different channels and modulations.

**Figure 15 sensors-22-05703-f015:**
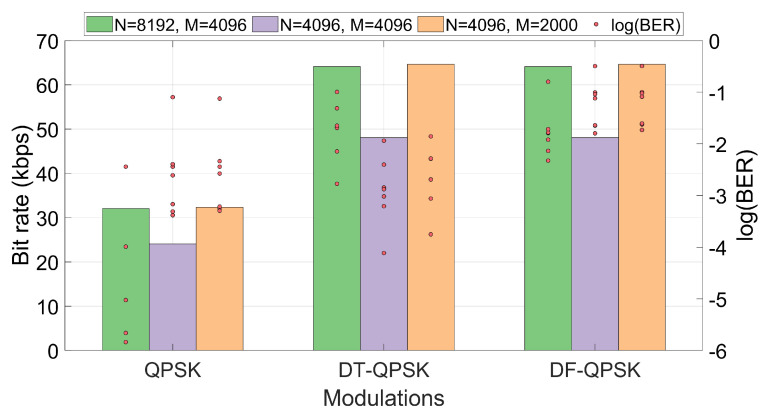
Comparison of the different proposed modulations based on the bit rate vs. BER tradeoff. The bars represent the bit rate, whose colors correspond to a given OFDM system configuration (*N* and *M* parameters). Eight dots representing the BER obtained in each of the eight channels are depicted on each bar.

**Figure 16 sensors-22-05703-f016:**
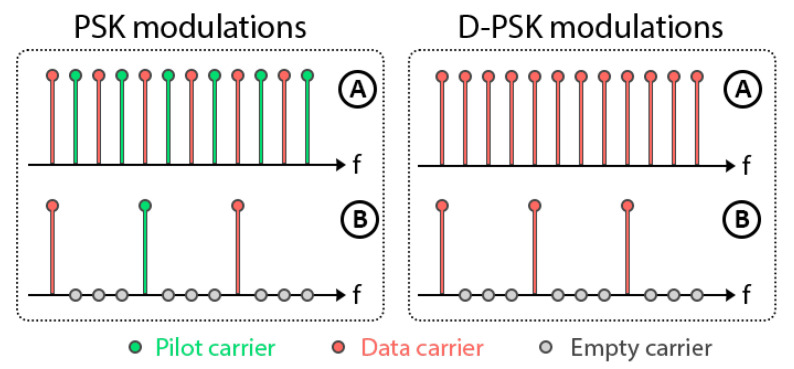
Simulated carrier patterns ((**A**) and (**B**)) for both PSK modulations, which make use of pilot carriers, and D-PSK modulations, in which all active carriers are used as data carriers.

**Figure 17 sensors-22-05703-f017:**
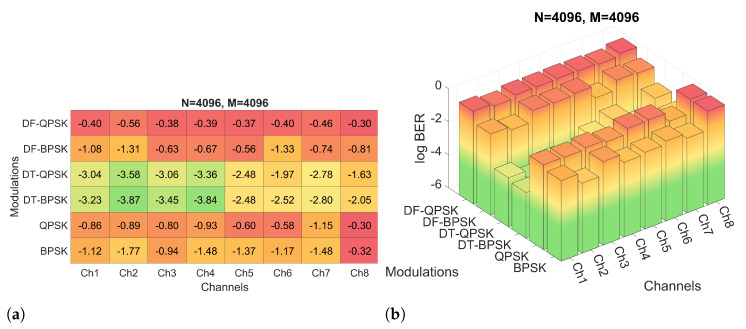
log BER for configuration 2 and carrier pattern B with different channels and modulations: (**a**) numerical heat matrix and (**b**) bar diagram.

**Figure 18 sensors-22-05703-f018:**
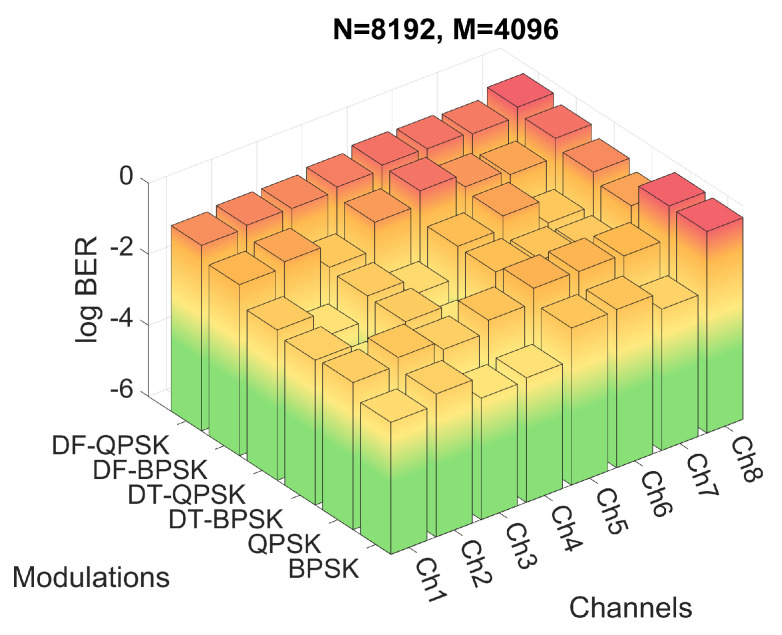
BER for configuration 1 and carrier pattern B with different channels and modulations.

**Figure 19 sensors-22-05703-f019:**
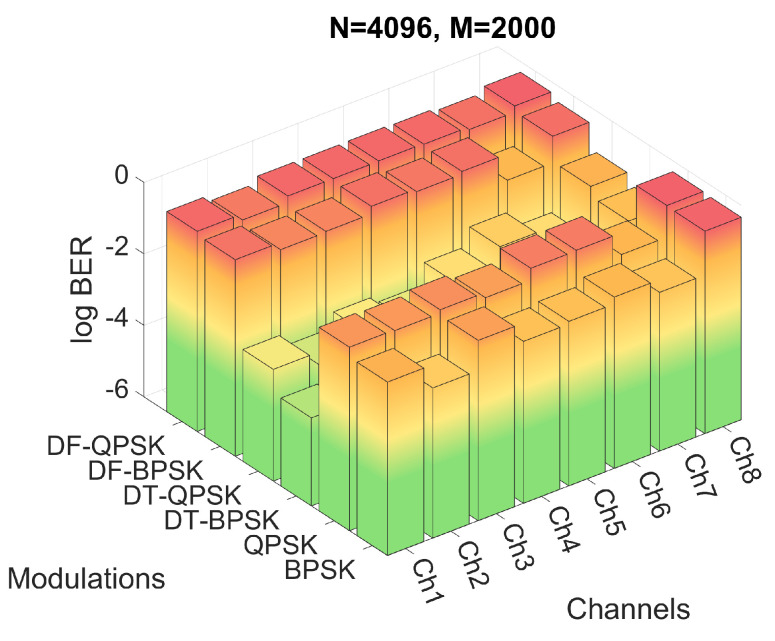
BER for configuration 3 and carrier pattern B with different channels and modulations.

**Figure 20 sensors-22-05703-f020:**
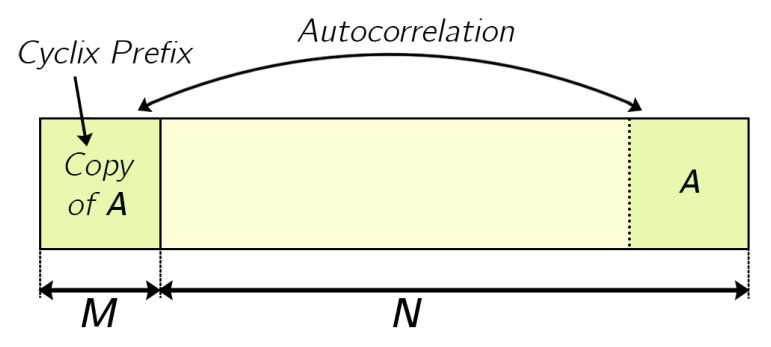
Autocorrelation of the CP with its copy in a transmitted OFDM symbol.

**Figure 21 sensors-22-05703-f021:**
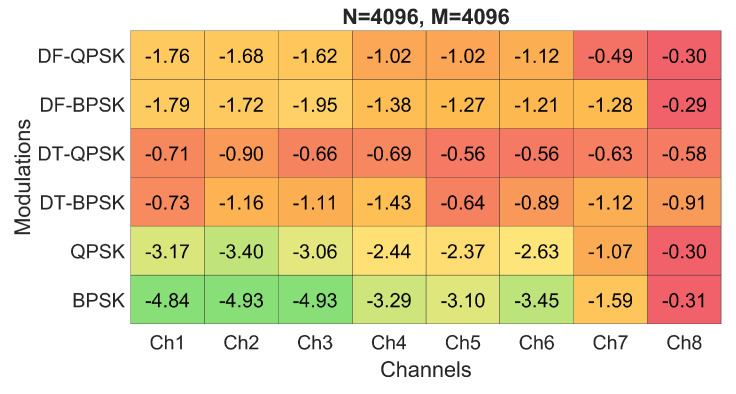
BER for configuration 2 and carrier pattern A with different channels and modulations and using a simplified CP correlation-based synchronization technique.

**Table 1 sensors-22-05703-t001:** Estimated transmitter–receiver separation for the eight measured channels.

Channel	1	2	3	4	5	6	7	8
Separation (m)	243	102	128	228	239	252	258	97

**Table 2 sensors-22-05703-t002:** Set of values (or range of them) to the OFDM system parameters.

Number of FFT points	*N*	8192,4096
Number of active carriers	$N_{a}$	1576, 788
Number of samples in CP	*M*	4096, 2000
Sample frequency	$f_{s}$	500 kHz
Center frequency	$f_{c}$	80 kHz
Bandwidth	BW	96 kHz
Bandwidth used	-	(32–128) kHz

**Table 3 sensors-22-05703-t003:** Preselected OFDM parameter settings for performance analysis.

Configuration Number	1	2	3
Number points in FFT (*N*)	8192	4096	4096
Number of samples in CP (*M*)	4096	4096	2000

**Table 4 sensors-22-05703-t004:** Channel variations within the same OFDM symbol.

Configuration Number	1	2	3
Symbol duration (ms)	25.58	16.38	12.19
Intra-symbol channel changes	5.85	3.90	2.90

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
