# Peer review of "OFDM System Design for Measured Ultrasonic Underwater Channels"

_sensors, 2022, doi:10.3390/s22155703_

Round 1

Reviewer 1 Report

The paper is devoted to an interesting topic. It is well-structured and well-written. The content presentation is easy to follow.

Although the results are interesting and can help with the further synthesis of UAC systems, some minor issues are present.

Some minor grammar and punctuation issues are present (e.g., in line 96 “to kinds of …” should presumably be “two kinds of …”, in line 103 the word selection “choice …obeys” seems to be inappropriate, etc.) and must be fixed in the final version. 

Some abbreviations are introduced without being defined in the text at their first appearance (e.g., ICI).

The authors suggested restructuring sections 3.1 and 3.2 (or merging them) since it is hard to detach them as separate ones (both deal with the OFDM signal synthesis).

From the text, it is not clear enough the difference between the measured channels. Is it only separation distance? If yes, then why they were not ordered in a distance-increasing manner? This will help to follow the results easier.   

Fig. 15  is an essential element of the research (since it helps to find a compromise between Rate and BER), but it is hard to read it, and the text doesn’t bring any explanation. It is not clear what the Rb is? Was it measured for some specific channel, or it was aggregated somehow from all of those channels (if they were all active simultaneously)?

In Tab.4 the parameter “intra-symbol channel changes” is not defined or quantified.

In Section 4.3 classical CP-OFDM is described (without any innovations) thus, there is no need for such a detailed description. At the same time, very little space is devoted to the description of the results. It is expected that the discussion of the results will be expanded.

Reviewer 2 Report

Please find enclosed my report.
